# Assessing perceived and functional health literacy among parents in Cyprus: A cross-sectional study

Joanna Menikou [1,2]*, Nicos Middleton[1], Evridiki Papastavrou[1], Christiana Nicolaou[1]

1 Department of Nursing, Cyprus University of Technology, Limassol, Cyprus, 2 Department of Nursing, European University of Cyprus, Nicosia, Cyprus

☯ These authors contributed equally to this work.
* i.menoikou@edu.cut.ac.cy

## Abstract

**Data Availability Statement:** All relevant data are within the paper and as Supporting Information files. In specific, all tools used for data collection are attached as Supporting Information.

### Background

Parents often must take decisions regarding their children's health, which requires certain skills and competences. Parental health literacy (HL) is important in establishing positive health-promoting behaviours and better health outcomes to their children. Limited parental HL has been associated with various negative children's health outcomes. The aim of the study was to explore perceived and functional HL among parents in Cyprus.

### Method

A cross-sectional study was conducted with a convenience sample of 416 parents of children, aged 6 months to 15 years old, presenting in pediatric outpatient departments across three Cypriot cities. Participants completed the HLS-EU-Q47, a self-reported measure of HL, and the NVS (Newest Vital Sign), a performance-based measure of HL. Associations with socio-demographic characteristics and health behaviors were explored.

### Results

Based on suggested ranges, among 416 parents, mostly mothers (83.2%), almost half of parents (42.6%) were classified as having inadequate or problematic perceived HL. Consistently, 62.8% showed high likelihood or significant possibility of limited functional HL, based on the NVS with a mean score of 2.73 out of 6 (SD = 2.02). Nevertheless, no correlation was observed between the two measures of HL. Limited parental perceived HL was statistically significantly associated with lower educational attainment, lower number of children in the family, increased self-assessed health status, and limited exercise habits. Limited parental functional HL was statistically significantly associated with female gender, younger age, lower educational attainment, receiving financial aid, lower family income, and lower alcohol consumption.

Furthermore, the study's minimal data set has been published - https://doi.org/10.6084/m9.figshare.23807673.v1

**Funding:** The authors received no specific funding for this work.

**Competing interests:** The authors have declared that no competing interests exist.

## Conclusion

Even though there was lack of agreement in the classification according to the HLS-EU-Q47 and the NVS, moderate-to-low levels of perceived HL appear consistent with a performance-based measure of HL. As a high number of parents may face challenges in assessing and applying health information to improve outcomes for their children, with evidence of social gradient, healthcare services should be oriented towards identifying problematic HL while Public Health interventions are needed to enhance parental HL.

## Introduction

Health literacy (HL) has been recognized as the 'sixth vital sign' [1], and low levels of HL have been acknowledged as 'silence epidemic' [2]. Recently, during the COVID-19 pandemic, HL has been characterized as a 'social vaccine' able to influence individual and public health decisions and actions [3]. It is well established that HL had a significant role in public health. Policies, strategies, and action plans have been developed at a local, national, and global level to enhance populational HL by encouraging healthy lifestyles, preventing diseases, promoting health, and managing illnesses [4–6].

Parental HL seems to directly impact children's health status. Parents constantly have to deal with health information and services, provide health care, and take decisions about their children's health issues, to maintain their child's health and well-being. Parents often need the competencies to find and assess reliable health information for a variety of children's health issues, to effectively communicate with healthcare professionals, manage health conditions of their child, respond to symptoms, and deal with acute illness. Previous studies have linked limited parental HL with negative health behaviors and outcomes in children such as, increased likelihood of medication errors [7,8], poor Body Mass Index (BMI) [9], and limited knowledge or inappropriate practices regarding fever and its management [10]. In addition, two earlier systematic reviews [11,12] and a more recent scoping review [13] revealed that limited parental HL was associated with negative health behaviors and health outcomes in children. Therefore, a primary issue in promoting health, preventing illness, and providing effective healthcare in children could be the enhancement of parental HL.

HL is a broadly defined term. In a systematic review [14], 250 different HL definitions were identified. It can be observed that HL is a multidimensional concept consisting by a variety of components, such as reading, numeracy, and cognitive skills and behaviors that an individual must have to function effectively within a healthcare system. A more comprehensive definition of HL derived from the review of 17 conceptual frameworks and definitions [15] is that HL "entails people's knowledge, motivation and competences to access, understand, appraise, and apply health information in order to make judgments and take decisions in everyday life concerning healthcare, disease prevention and health promotion to maintain or improve quality of life during the life course".

The diversity which exists in the conceptual models of HL led to the development of a variety of tools to measure different aspects of HL. Some of HL tools are shorter and limited to identify difficulties in reading and/or numeracy skills, and others are broader measuring more complex HL skills such as understanding and appraising health information [16]. Moreover, some HL tools are performance-based measurements and assess functional HL taking into consideration the skills that a person needs in health context such as reading and numeracy skills [17], whereas others are self-reported and measure an individual's perceived HL in

broader elements of HL such as, personal and interpersonal capacities, health system aspects, and community factors. According to a search on PubMed, the three most mentioned or cited HL tools per year were TOFHLA (Test of Functional Health Literacy in Adults) [18] and NVS (Newest Vital Sign) [16,19] among functional HL tools and HLS-EU (European Health Literacy Survey) [20] among perceived HL measurement tools.

The great majority of previous studies on HL have commonly used a single tool to measure HL focusing either on self-assessed HL or performance-based HL [21–23]. Few studies have explored in parallel both self-assessed (perceived) HL and performance-based (functional) HL. A systematic review [24] identified only four fair quality studies which explored the association between both performance-based and self-assessed HL and health outcomes with mixed results. Nevertheless, most studies included in the systematic review found a similar relationship between results of performance-based and self-assessed tools and health outcomes. Several studies since then have explored both performance-based and self-assessed HL but they still remain limited. They either measured different types of HL, such as self-assessed tool to measure eHealth Literacy, which refers to the ability to seek, find, understand, and appraise health information from electronic sources [25], and performance-based tool to measure HL [26–28]. Some others used two performance-based tools [29], or a single screening question as the second HL tool [30]. Furthermore, those studies were conducted among varied population groups other than parents.

Parents are the ones who are responsible for their children's health. Identifying parents with limited HL is vital in ensuring best practices regarding health promotion, disease prevention, and healthcare in childhood population. Furthermore, assessing both types of HL (self-assessed and performance-based) may lead in a more holistic and comprehensive assessment of parental HL. Thus, the primary aim of the current study was to examine both self-assessed HL and performance-based HL among Cypriot parents. Several sociodemographic characteristics have been associated with limited HL in adult population in terms of gender [31], age [26,32], nationality [33–35], and educational attainment [26,32,37]. Therefore, an additional objective of the current study was to identify a set of characteristics of a parent with limited HL. Hence, the following research questions were examined:

1. Are sociodemographic characteristics (such as educational attainment, financial status) potential predictors of parental perceived HL and functional HL?

2. Are health-related behaviours (such as smoking, exercise habits) potential predictors of parental perceived HL and functional HL?'

## Materials and methods

### Study design and sample

This is a descriptive study with a correlational design. Study participants were recruited over a 2-year period, between January 2020 and December 2021 from three paediatric primary care practices across three different Cypriot cities; Nicosia, Limassol, and Paphos. While a convenience sampling approach was employed, the inclusion of three primary care practices aimed at capturing enough variation of the study population in terms of sociodemographic characteristics. Since the recent establishment of the General Health System (GeSY), the paediatric population has direct access to a paediatrician, with no co-payments. Like General Practitioners, paediatricians get reimbursed by GeSY based on list size rather than number of visits. With no geographical restrictions, families can register with any paediatrician of their choice. This effectively reduced previous socio-economic disparities when healthcare expenditure was covered by private insurance and/or out-of-pocket payments.

The inclusion criteria were: (a) parent or guardian accompanying the child to the practice, (b) older than 18 years of age, irrespective of gender (e.g. mother or father of the accompanying child), (c) the child in reference was aged six months to fifteen years old without any chronic medical condition or a history of a premature birth, (d) permanent resident of Cyprus and GeSY recipients, and (e) ability to communicate, read and write in either Greek or English to complete the questionnaire pack. Grandparents or other individuals with no parental legal responsibility accompanying the child to the primary care practice were not included.

The minimum sample size was set to 369. This was based on precision analysis using estimates from a previous study in the Cypriot population which used the HLS-EU-Q47 [36]. In that study, the percentage of healthcare users aged 30–44 years old with inadequate or problematic HL was estimated at 39.6%, while the observed standard deviation in HL scores was in the magnitude of 8. Thus, precision analysis suggests that the sample size allows the estimation of 95% confidence intervals for proportions with ±5% statistical error in the full sample.

### Data collection procedure

Recruitment of study participants was performed solely by the main researcher (JM) who is a Registered Nurse with no employment or other connection with the specific primary care practices. Visits were performed 2–3 times per week and at different times a day in consecutive time periods for each practice. Recruitment of potential participants was performed according to a study protocol, detailing the eligibility criteria for inclusion in the study, data collection process (i.e. data collection at different days and times each week) and sampling. In terms of sampling, a consecutive sample of practice visitors who fulfilled the eligibility criteria were approached and invited to participate in the study. A pilot study was also undertaken prior to the main study to explore both the feasibility of the study protocol as well as the measurement reliability of the questionnaire in the specific population.

After informing eligible parents about the scope and process of the study, including a description of the tools and tasks that will be required from them, enrolled participants provided informed consent in writing. Participation was voluntary, while participants were informed that they could withdraw their participation at any stage in the process. A printed questionnaire pack (sociodemographic, health-behaviours, and HLS-EU-Q47) was provided. Parents completed the pack while waiting for their appointment, in the main waiting room or a separate private room where possible, if this was their preference. Parents were also given a hard copy of the NVS nutrition label to review and refer to, while the main researcher asked the six questions of the scenario. If both parents were present, the opportunity to decide who would participate in the study was provided, but only one parent per child was included in the sample. The completed questionnaires were returned anonymously, to ensure that no one could identify individual participants after data collection, to the main researcher before leaving the primary care practice. Ethics approval was obtained from the National Bioethics Committee of Cyprus (ref.: EEBK ΕΠ 2018.01.205) as well as the Research Promotion Committee of the Cyprus Ministry of Health.

### Assessment of outcome variables

#### Perceived HL

Perceived HL was assessed using the Greek-version of the HLS-EU-Q47 questionnaire. This is one of the most widely used measures of HL, originally developed in the context of a European project across eight countries, namely Austria, Bulgaria, Germany, Greece, Ireland, Netherlands, Poland, and Spain [37]. The tool consists of 47 items measuring perceived HL with regard to three domains: healthcare (16 questions), disease prevention (15 questions), and

health promotion (16 questions). In all domains, items tap on four competences that influence a person's decision making: accessing, understanding, appraising, and applying health information. The answers are given on a four-point Likert response scale whereby 1 corresponds to "very difficult" to 4 to "very easy". The total score is expressed in the 0 to 50 range, whereby score <26 indicates inadequate, 26–33 problematic, 34–42 adequate HL, and 43–50 excellent perceived HL. The tool has been shown to have good reliability and validity [22,38–40]. In regard to the Cypriot population, a pilot validity assessment of the Greek translated version of the tool showed good metric properties [36]. Specifically, the study supported the construct factor validity of the tool. Furthermore, the study showed that the tool was able to capture the expected social gradient in HL by indicators of socio-economic disadvantage, including educational attainment, further supporting its criterion-based validity. In terms of the dimensionality, the recommendation was to 4 skills sub-scales or 3 health domain sub-scales, rather than across all 12 sub-scales of the 4X3 HL conceptual model. As this is not an uncommon finding in the literature, this conceptualization was preferred here to calculate aggregate scores, only for descriptive purposes. Only the overall score, with a high level of internal consistency, as determined by a Cronbach's a of 0.958, was used in the analysis.

**Functional HL.**   Functional HL was assessed using the NVS tool. The NVS was developed by Pfizer Inc in English and Spanish to measure functional HL. It is a task-based measure of HL based on an ice cream nutrition label with six related questions assessing prose literacy, document literacy, and numeracy skills. The total score ranges from 0 to 6. A score of 0 and 1 suggests high likelihood of limited HL, score of 2 and 3 suggests the possibility of limited HL, and score between 4 and 6 suggests adequate HL. The tool was reported to have good internal consistency with Cronbach a >0.76 and has been validated against other tools that measure functional HL, such as TOFHLA [19].

## Assessment of predictor variables

**Sociodemographic and health behaviours.**   Participants were also asked to provide basic sociodemographic information, including age, gender, marital status, nationality, place of residence (urban vs rural), and other, expected to demonstrate an association with HL based on previous studies [36,37]. Furthermore, in order to explore the potential social gradient in parental HL and in the absence of a standard measure of socioeconomic status in Cyprus, a range of variables was explored including educational attainment, monthly family income, employment status, financial difficulties (e.g. paying bills in the past 12 months), and/or receipt of financial aid. Finally, the subjective perception of relative standing on the social hierarchy was measured using a variation of the MacArthur Scale [41]. Such measures are considered particularly relevant as they represent an internalized perception of social position through a process of social comparison. Specifically, the measure represents the social structure as a ladder with 10 steps, where responders locate themselves on the ladder considering that the top represents people who are best-off (most privileged in terms of education, income, and employment) and the bottom people who are worst-off (most disadvantaged).

Parents also provided information on a range of health-related behaviours, such as smoking habits, alcohol consumption, and provided a standard self-assessment of their own health status on a five-point scale ranging from poor to excellent.

## Statistical analysis

Descriptive statistics, including frequency (n), percentage (%), mean (M), and standard deviation (SD) were used to describe parental sociodemographic characteristics, health behaviors, perceived HL, and functional HL. All variables of sociodemographic characteristics, and health

behaviors were treated as categorical variables, except from smoking pack-years. Furthermore, Pearson correlation was used to determine the strength and direction of a linear relationship between HLS-EU-Q47 score and NVS score.

Pack years was addressed as the total measurement which combines the number of cigarettes that a person smokes on a daily basis (in a packet of 20 cigarettes) with the total period that the person smokes. For example, one pack year is one packet of cigarette for one year, or half of packet of cigarette for two years etc. Pack years were calculated for current smokers and post smokers.

The outcome variables of perceived HL and functional HL were operationalized as both categorical and continuous variables. To examine bivariable associations between the continuous outcome variables (perceived HL and functional HL) and predictor variables (sociodemographic characteristics, health behaviors), a series of independent sample t-tests and one-way Analysis of Variance (ANOVAs) were conducted. To examine whether there was a relationship between outcome variables (perceived HL, functional HL) treated as categorical variables, and predictor variables, chi-square tests for independence ($\chi^2$) were used. Games-Howell post-hoc analysis or Tukey post-hoc analysis was conducted to examine whether the differences between groups were statistically significant.

Stepwise multiple linear regression analyses were used to identify the minimum set of variables associated with parental perceived HL or functional HL scores with all predictor variables: parental sociodemographic characteristics and parental health behaviors after mutually adjusting for each other. The strength of the association between predictor variables and dependent variables was assessed by computing standardized regression coefficients (tiny ≤0.05, very small 0.05 to 0.10, small 0.10 to 0.20, medium 0.20 to 0.30, large 0.30 to 0.40, and very large ≥0.40) [42] and Cramer's V (weak from 0.10 to 0.30, medium from 0.40 to 0.50, and strong >0.50) as effect sizes. Statistical significance was defined as $p < 0.05$. All statistical analysis were performed using SPSS version 26.

## Results

### Descriptive analyses

**Sociodemographic data.** The majority among 416 participating parents were female (n = 346, 83.2%), married or in partnership (n = 358, 86.1%), of Cypriot nationality (n = 380, 91.3%) and resident in an urban area (n = 285, 68.5%). Regarding parental age, almost one in three parents (n = 129, 31.0%) were 30–34 years old. Almost half of them had two children (n = 193, 46.4%), and seven to ten parents were working full time (n = 290, 69.7%). The monthly family income in half of participants was under 2400 euro (n = 210, 50.4%). Almost seven in ten reported no financial difficulties (n = 285, 68.5%) and about half (n = 219, 52.6%) reported that they did not receive any financial allowance. Finally, in terms of subjective social status, 14.2% of participants placed themselves at the bottom steps (1–4) of the ladder and only 10.8% on the top steps (8–10). The majority perceived themselves being between the sixth and seventh step, indicating middle-to-high social status (n = 177, 42.6%). Finally, a high percentage of participants were University graduates with a bachelor or postgraduate degree (n = 286, 68.7%). The socio-demographic profile of participants is presented in Table 1.

**Health behaviours.** Most parents self-rated their health as good (n = 197, 47.4%) or very good (n = 129, 31.0%). As many as half reported that they visited the doctor for themselves only once or twice in the past year (n = 219, 52.6%), while 19.7% reported that they never visited the doctor. About half of them reported that they have never smoked (n = 226, 54.3%) and did not consume any alcoholic drink in the last month (n = 225, 54.1%). In the case of parents who were smokers (n = 92, 22.1%) the mean pack-years of smoking was 9.65 (SD = 11.62),

**Table 1. Descriptive analyses of sociodemographic data and health behaviours of the participants (n = 416).**

| Variable | Frequency (n) | Percentage (%) |
|---|---|---|
| **Sociodemographic characteristics** | | |
| **Gender** | | |
| Male | 70 | 16.8 |
| Female | 346 | 83.2 |
| **Age group** | | |
| 18–29 | 79 | 19.1 |
| 30–34 | 129 | 31.0 |
| 35–39 | 104 | 25.0 |
| ≥40 | 104 | 24.9 |
| **Place of residence** | | |
| Urban area | 285 | 68.5 |
| Rural area | 113 | 27.2 |
| **Nationality** | | |
| Cypriot | 380 | 91.3 |
| Other | 36 | 8.7 |
| **Marital status** | | |
| Married/In partnership | 358 | 86.1 |
| Not married/Divorced | 43 | 10.4 |
| **Educational attainment** | | |
| At most Secondary | 51 | 12.2 |
| College | 63 | 15.1 |
| University | 159 | 38.2 |
| Postgraduate | 127 | 30.5 |
| **Financial aid** | | |
| Yes | 179 | 43.0 |
| No | 219 | 52.6 |
| **Financial difficulties** | | |
| Yes | 113 | 27.2 |
| No | 285 | 68.5 |
| **Number of children** | | |
| 1 | 148 | 35.6 |
| 2 | 193 | 46.4 |
| ≥3 | 60 | 14.4 |
| **Monthly family income** | | |
| ≤1849 | 147 | 35.3 |
| 1850–2949 | 125 | 30.0 |
| 2950–4399 | 73 | 17.6 |
| ≥4400 | 50 | 11.9 |
| **Employment status** | | |
| Full time | 290 | 69.7 |
| Part time | 41 | 9.9 |
| Unemployed | 66 | 15.7 |
| **Perceived social status** | | |
| 1–4 level | 59 | 14.2 |
| 5 level | 113 | 27.2 |
| 6–7 level | 177 | 42.6 |
| 8–10 level | 45 | 10.9 |

(*Continued*)

**Table 1.** (Continued)

| Variable | Frequency (n) | Percentage (%) |
|---|---|---|
| **Health behaviors** | | |
| **Self-rated health** | | |
| Very good | 129 | 31.0 |
| Good | 197 | 47.4 |
| Less than good Moderate | 72 | 17.3 |
| **Doctor visits during the last 12 months** | | |
| None | 87 | 20.9 |
| 1–2 times | 219 | 52.6 |
| 3–5 times | 54 | 13.0 |
| ≥6 times | 39 | 9.4 |
| **Smoking habits** | | |
| Smoking | 92 | 22.1 |
| Used to smoke, but stopped | 81 | 19.5 |
| Never smoked | 226 | 54.3 |
| **Smokers pack years** | 9.65 (11.62) | |
| **Past smokers pack years** | 6.21 (6.76) | |
| **Alcohol consumption during the last month** | | |
| Yes | 174 | 41.8 |
| No | 225 | 54.1 |
| **Frequency of alcohol consumption** | | |
| >1 times/week | 23 | 13.0 |
| 1 time/week | 47 | 26.7 |
| 2–3 times/month | 50 | 28.4 |
| 1 time/month | 56 | 31.8 |
| **Exercise habits** | | |
| Almost every day | 40 | 9.6 |
| Sometimes per week | 101 | 24.3 |
| Sometimes per month | 83 | 20.0 |
| Never | 172 | 42.0 |

which corresponds with one packet of cigarettes for more than nine years, while for past smokers, the mean pack-years of smoking was 6.21 (SD = 6.76) which corresponds with one packet of cigarettes for almost seven years. Finally, in terms of exercise habits, two in five parents did not exercise at all (n = 172, 42.0%). The self-reported health-behaviors are also presented in Table 1.

**Perceived and functional HL.** Based on the HLS-EU-Q47 classification, only 7.0% (n = 29) of parents demonstrated inadequate HL levels, however, a further 35.6% (n = 148) of parents self-classified under problematic HL. The mean score of perceived HL was 35.30 (SD = 7.45) (Table 2).

Among the four skills of HL, the dimension with the highest score of HL was understanding (M = 37.71, SD = 7.39), followed by applying (M = 35.60, SD = 8.01), and access (M = 34.63, SD = 8.34). The lowest score was observed for appraising health information (M = 33.55, SD = 9.05). In terms of domains, the highest total HL score was observed for the healthcare domain (M = 36.16, SD = 7.04), followed by disease prevention (M = 35.15, SD = 8.87), and health promotion (M = 34.60, SD = 8.88) (Table 2).

According to a detailed analysis across all HLS-EU-Q47 items, following instructions from doctor/pharmacist and following instructions from medicine leaflets were perceived as the

**Table 2. Descriptive analyses of outcome variables (perceived and functional HL) of the participants (n = 416).**

| Outcome variables | | |
|---|---|---|
| **Variable** | **Frequency (n)** | **Percentage (%)** |
| **HLS-EU-Q47** | 35.30(7.45) | |
| Inadequate (<26 score) | 29 | 7.0 |
| Problematic (26–33 score) | 148 | 35.6 |
| Adequate (34–42 score) | 142 | 34.1 |
| Excellent (43–50 score) | 97 | 23.3 |
| *Skills* | | |
| Accessing | 34.63(8.34) | |
| Understanding | 37.71(7.39) | |
| Appraising | 33.55(9.05) | |
| Applying | 35.60(8.01) | |
| *Health Domains* | | |
| Healthcare | 36.16(7.04) | |
| Disease Prevention | 35.15(8.87) | |
| Health Promotion | 34.60(8.88) | |
| **NVS** | 2.73(2.02) | |
| Adequate HL (4–6 score) | 140 | 37.2 |
| Possibility of limited HL (2–3 score) | 106 | 28.2 |
| High likelihood of limited HL (0–1 score) | 130 | 34.6 |

easiest tasks, among 99.8% (n = 415) and 99.3% (n = 413) respectively, whereas finding out about political changes that may affect health and judging if the information about illness in the media is reliable were considered as the most difficult, with 51.7% (n = 215) and 50.5% (n = 210) respectively–results not shown in detail.

The mean score of functional HL was 2.73 (SD = 2.02), indicating a possible limited HL. In fact, according to the NVS classification, 34.6% (n = 130) of parents were identified as high likelihood and 28.2% (n = 106) as significant possibility of limited functional HL (Table 2). Parents demonstrated more difficulty in items 4 ('If you usually eat 2.500 calories in a day, what percentage of your daily value of calories will you be eating if you eat one serving?) (n = 255, 61.3%) and 3 ('Your doctor advises you to reduce the amount of saturated fat in your diet. You usually have 42g of saturated fat each day, which includes one serving of ice cream. If you stop eating ice cream, how many grams of saturated fat would you be consuming each day?) (n = 212, 51.0%) which require more advanced numeracy skills, such as mathematical operations using proliferation or division. However, although questions 1 and 2 require basic numeracy (mathematical operations using addition or subtraction) and reading skills, the number of parents who had difficulty in replying to them was also increased, with 47.8% (n = 199) and 41.3% (n = 172), respectively. Interestingly, the number of wrong answers was also high in question 5 (n = 170, 40.9%) which required only basic reading and understanding skills–results not shown in detail.

Perhaps, surprisingly, no correlation was observed between perceived HL (HLS-EU-Q47 score) and functional HL (NVS score), $r(374) = -0.03$, $p = 0.50$. Interestingly, the highest score of HLS-EU-Q47 (M = 36.37, SD = 7.52) was observed among the group of parents with high likelihood of limited HL according to the NVS. Similarly, the lowest NVS scores were observed among parents classified at opposite ends of the HLS-EU-Q47 scale with the lowest NVS score (M = 2.47, SD = 2.05) recorded among the self-reported excellent HL according to the HLS-EU-Q47. Table 3 presents a cross-tabulation and mean scores of the two measures of HL.

**Table 3. Cross-tab and mean scores of outcomes (perceived and functional HL).**

| Outcome variables | | NVS categories | | | | NVS score M(SD) |
|---|---|---|---|---|---|---|
| | | High likelihood of limited HL (0–1 score) % (n) | Possibility of limited HL (2–3 score) % (n) | Adequate HL (4–6 score) % (n) | Overall % (n) | |
| HLS-EU-Q47 categories | Inadequate (<26 score) | 1.9% (7) | 2.9% (11) | 2.4% (9) | 7.2% (27) | 2.59 (1.76) |
| | Problematic (26–33 score) | 10.4% (39) | 12.0% (45) | 14.6% (55) | 37.0% (139) | 2.89 (2.06) |
| | Adequate (34–42 score) | 12.0% (45) | 8.5% (32) | 12.0% (45) | 32.4% (122) | 2.77 (2.03) |
| | Excellent (43–50 score) | 10.4% (39) | 4.8% (18) | 8.2% (31) | 23.4% (88) | 2.47 (2.05) |
| | Overall % (n) | 34.6% (130) | 28.2% (106) | 37.2% (140) | | |
| HLS-EU-Q47 score M(SD) | | 36.37 (7.52) | 33.72 (7.62) | 35.11 (7.34) | 100% (376) | |

The lack of agreement in the classification according to the two measures of HL is apparent. Characteristically, as many as one in three parents were classified as having high likelihood of limited HL according to the NVS (n = 130, 34.6%). However, among this group of parents, the majority appeared to have adequate perceived HL (n = 45, 34.6%) or excellent (n = 39, 30.0%) perceived HL. The low observed correlation between perceived HL and functional HL might suggest that concurrent validity is limited between the two tools (NVS and HLS-EU-Q47). However, subjective assessment of HL (HLS-EU-Q47) may not necessarily be indicative of the actual performance in a specific HL-related task due to the restricted range of HL skills covered by the NVS (i.e nutritional label).

## Bivariate associations of parental characteristics and behaviours with perceived HL and functional HL

**Perceived HL.** The results of bivariate analyses of perceived HL, expressed either as a continuous or as categorical variable, with socio-demographic characteristics and health-related behaviors are presented in Tables 4 and 5 respectively. A statistically significant association with perceived HL was observed with the following characteristics: educational attainment ($F$ $(4,411) = 3.186$, $p = 0.01$) and number of children ($F(3,412) = 3.898$, $p = 0.01$)–see Table 4. Specifically, mean scores of parental perceived HL showed a stepwise pattern by educational attainment, increasing from 33.81 (SD = 6.56) among parents with at most secondary education (n = 51), to 36.14 (SD = 7.43) among college (n = 63), and 36.27 (SD = 7.47) among university graduates (n = 159). This corresponds to 43.1% of parents with at most secondary education self-assessing their HL as adequate or excellent versus over 60% among those with tertiary education (college or University). Interestingly, this pattern did not continue with the group of parents with postgraduate education (n = 127), where the mean score was 33.91 (SD = 7.59), and equivalently around 50% self-rating their HL as adequate or excellent. An increase in perceived HL score was also observed from 33.99 (SD = 7.00) in parents with a single child (n = 148) to 36.18 (SD = 7.76) in parents with two children, which was statistically significant ($p = 0.01$).

With regard to health-related behaviours (see Table 5), self-rated health status ($F(3,412) =$ $3.962$, $p = 0.01$), and exercise habits ($\chi^2(4) = 10.482$, $p = 0.03$) were associated with perceived HL. Specifically, perceived parental HL appears higher on average among parents who self-rated their health as very good/excellent (n = 129) at 36.87 (SD = 7.54) compared to those who

**Table 4. HLS-EU-Q47 and NVS mean scores and categories according to parental socio-demographic characteristics.**

| | HLS-EU-Q47 | | | | | NVS | | | | | |
| | Inadequate/ Problematic n (%) | Adequate/ Excellent n (%) | p-value† | Mean (SD) | p-value | High likelihood of limited HL n (%) | Possible limited HL n (%) | Adequate HL n (%) | p-value | Mean (SD) | p-value |
|---|---|---|---|---|---|---|---|---|---|---|---|
| **Gender** | | | | | | | | | | | |
| Male | 32(45.7) | 38(54.3) | 0.56 | 34.35 (7.57) | 0.24 | 21(34.4) | 10(16.4) | 30(49.2) | 0.04 | 3.08 (2.25) | 0.18 |
| Female | 145(41.9) | 201(58.1) | | 35.50 (7.42) | | 109(34.6) | 96(30.5) | 110(34.9) | | 2.66 (1.97) | |
| **Age group** | | | | | | | | | | | |
| 18–29 | 27(34.2) | 52(65.8) | 0.36 | 36.17 (7.01) | 0.24 | 27(39.7) | 21(30.9) | 20(29.4) | 0.03†† | 2.37 (1.85) | 0.04 |
| 30–34 | 56(43.4) | 73(56.6) | | 34.70 (7.22) | | 46(38.7) | 35(29.4) | 38(31.9) | | 2.47 (1.92) | |
| 35–39 | 45(43.3) | 59(56.7) | | 36.12 (7.88) | | 26(27.4) | 29(30.5) | 40(42.1) | | 3.07 (2.09) | |
| ≥40 | 49(47.1) | 55(52.9) | | 34.58 (7.57) | | 31(33.0) | 21(22.3) | 42(44.7) | | 2.98 (2.14) | |
| **Place of residence** | | | | | | | | | | | |
| Urban area | 123(43.2) | 162(56.8) | 0.72 | 34.95 (7.31) | 0.40 | 90(33.5) | 76(28.3) | 103(38.3) | 0.79 | 2.80 (2.00) | 0.33 |
| Rural area | 51(45.1) | 62(54.9) | | 35.65 (7.83) | | 39(36.8) | 30(28.3) | 37(34.9) | | 2.58 (2.08) | |
| **Nationality** | | | | | | | | | | | |
| Cypriot | 158(41.6) | 222(58.4) | 0.19 | 35.39 (7.46) | 0.45 | 121(35.4) | 96(28.1) | 125(36.5) | 0.55 | 2.68 (2.01) | 0.11 |
| Other | 19(52.8) | 17(57.5) | | 34.42 (7.37) | | 9(11.8) | 10(29.4) | 15(44.1) | | 3.26 (2.11) | |
| **Marital status** | | | | | | | | | | | |
| Married/In partnership | 156(43.6) | 202(56.4) | 0.07 | 35.04 (7.52) | 0.16 | 110(32.8) | 98(29.3) | 127(37.9) | 0.12 | 2.75 (2.00) | 0.57 |
| Not married/ Divorced | 19(44.2) | 24(55.8) | | 36.54 (6.88) | | 20(48.8) | 8(19.5) | 13(31.7) | | 2.56 (2.21) | |
| **Educational attainment** | | | | | | | | | | | |
| At most secondary | 29(56.9) | 22(43.1) | <0.01 | 33.81 (6.56) | 0.01 | 26(57.8) | 9(20.0) | 10(22.2) | <0.01†† | 1.91 (1.83) | <0.01 |
| College | 24(38.1) | 39(61.9) | | 36.14 (7.43) | | 25(43.9) | 17(29.8) | 15(26.3) | | 2.28 (2.05) | |
| University | 59(37.1) | 100(62.9) | | 36.27 (7.47) | | 54(35.8) | 42(27.8) | 55(36.4) | | 2.65 (2.01) | |
| Postgraduate | 63(49.6) | 64(50.4) | | 33.91 (7.59) | | 25(20.3) | 38(30.9) | 60(48.8) | | 3.34 (1.94) | |
| **Financial aid** | | | | | | | | | | | |
| Yes | 82(45.8) | 97(54.2) | 0.50 | 34.95 (7.01) | 0.58 | 64(37.6) | 55(32.4) | 51(30.0) | 0.03 | 2.55 (1.96) | 0.11 |
| No | 93(42.5) | 126(57.5) | | 35.37 (7.85) | | 66(32.0) | 51(24.8) | 89(43.2) | | 2.88 (2.07) | |
| **Financial difficulties** | | | | | | | | | | | |
| Yes | 52(46.0) | 61(54.0) | 0.56 | 35.81 (7.66) | 0.30 | 41(39.4) | 30(28.8) | 33(31.7) | 0.34 | 2.49 (1.99) | 0.15 |
| No | 122(42.8) | 163(57.2) | | 34.95 (7.40) | | 89(32.7) | 76(27.9) | 107(39.3) | | 2.82 (2.03) | |
| **Number of children in the family** | | | | | | | | | | | |

*(Continued)*

**Table 4.** (Continued)

| | HLS-EU-Q47 | | | | | NVS | | | | |
|---|---|---|---|---|---|---|---|---|---|---|
| | Inadequate/ Problematic n (%) | Adequate/ Excellent n (%) | p-value† | Mean (SD) | p-value | High likelihood of limited HL n (%) | Possible limited HL n (%) | Adequate HL n (%) | p-value | Mean (SD) | p-value |
| 1 | 72(48.6) | 76(51.4) | 0.01 | 33.99 (7.00) | 0.01 | 48(34.3) | 48(34.3) | 44(31.4) | 0.81†† | 2.56 (1.92) | 0.32 |
| 2 | 73(37.8) | 120(62.2) | | 36.18 (7.76) | | 63(35.2) | 42(23.5) | 74(41.3) | | 2.83 (2.10) | |
| ≥3 | 30(50.0) | 30(50.0) | | 34.77 (6.96) | | 18(32.1) | 16(28.6) | 22(39.3) | | 2.89 (2.05) | |
| **Monthly family income** | | | | | | | | | | | |
| ≤1849 | 60(40.8) | 87(59.2) | 0.45 | 35.51 (6.97) | 0.52 | 62(45.6) | 37(27.2) | 37(27.2) | 0.01†† | 2.25 (1.97) | <0.01 |
| 1850–2949 | 60(48.0) | 65(52.0) | | 34.94 (7.68) | | 41(33.9) | 36(29.8) | 44(36.4) | | 2.75 (2.03) | |
| 2950–4399 | 28(38.4) | 45(61.6) | | 35.69 (8.02) | | 17(23.6) | 18(25.0) | 37(51.4) | | 3.21 (1.98) | |
| ≥4400 | 23(52.3) | 21(47.7) | | 33.48 (7.68) | | 8(18.6) | 13(30.2) | 22(51.2) | | 3.53 (1.88) | |
| **Employment status** | | | | | | | | | | | |
| Full time | 126(43.4) | 164(56.6) | 0.10 | 35.11 (7.55) | 0.42 | 91(32.6) | 79(28.3) | 109(39.1) | 0.28 | 2.83 (2.02) | 0.06 |
| Part time | 20(48.8) | 21(51.2) | | 35.17 (7.91) | | 16(41.0) | 14(35.9) | 9(23.1) | | 2.00 (1.75) | |
| Unemployed | 28(42.4) | 38(57.6) | | 35.44 (6.99) | | 23(39.7) | 13(22.4) | 22(37.9) | | 2.74 (2.15) | |
| **Perceived social status** | | | | | | | | | | | |
| 1–4 level | 30(50.8) | 29(49.2) | 0.35 | 34.28 (7.24) | 0.14 | 19(34.5) | 14(25.5) | 22(40.0) | 0.96†† | 2.84 (2.13) | 0.88 |
| 5 level | 49(43.4) | 64(56.6) | | 34.83 (7.74) | | 39(36.1) | 31(28.7) | 38(35.2) | | 2.60 (1.94) | |
| 6–7 level | 80(45.2) | 97(54.8) | | 34.94 (7.39) | | 56(32.7) | 52(30.4) | 63(36.8) | | 2.77 (2.05) | |
| 8–10 level | 15(33.3) | 30(66.7) | | 37.46 (7.09) | | 16(38.1) | 9(21.4) | 17(40.5) | | 2.76 (2.07) | |

† Pearson chi-square.

††p value for trend (linear by linear association) as variable is ordinal categorical.

self-rated their health as good (n = 197, M = 34.37, SD = 7.21) and less than good (n = 72, M = 34.45, SD = 7.69), which was statistically significant ($p$ = 0.01). Finally, 67.5% of parents who reported exercising almost every day demonstrated adequate/excellent HL, whereas the equivalent proportion of parents among those who reported exercising sometimes per week, sometimes per month, or never was statistically significantly lower at 59.4%, 51.8%, and 53.7%, respectively ($p$ = 0.03). No meaningful associations were observed with smoking habits or alcohol consumption. In terms of doctor visits, it appeared that parents who never visited the doctor or visited the doctor more than 6 times in the past 12 months, self-rated their HL higher on average than the rest, but this difference was not statistically significant at the 5% level ($p$ = 0.06 and $p$ = 0.12 with the categorical and continuous outcome variable respectively).

**Functional HL.** The results of bivariate analyses with functional HL, as a continuous and categorical outcome, are also presented in Tables 4 and 5. In terms of socio-demographic characteristics, variables associated with higher functional HL were male gender ($p$ = 0.04), older

**Table 5. HLS-EU-Q47 and NVS mean scores and categories according to parental health and health-related behaviors.**

| Variables | HLS-EU-Q47 | | | | | NVS | | | | | |
|---|---|---|---|---|---|---|---|---|---|---|---|
| | Inadequate/ Problematic n (%) | Adequate/ Excellent n (%) | p-value | Mean (SD) | p-value | High likelihood of limited HL n (%) | Possible limited HL n (%) | Adequate HL n (%) | p-value | Mean (SD) | p-value |
| **Self-rated health status** | | | | | | | | | | | |
| Very good | 46(35.7) | 83(64.3) | 0.17† | 36.87 (7.54) | 0.01 | 47(39.2) | 35(29.2) | 38(31.7) | 0.41†† | 2.40 (1.88) | 0.07 |
| Good | 95(48.2) | 102(51.8) | | 34.37 (7.21) | | 58(30.7) | 53(28.0) | 78(41.3) | | 2.95 (2.07) | |
| Less than good | 33(45.8) | 39(54.2) | | 34.45 (7.69) | | 25(37.3) | 18(26.9) | 24(35.8) | | 2.72 (2.09) | |
| **Doctor visits during the last 12 months** | | | | | | | | | | | |
| None | 34(39.1) | 53(60.9) | 0.06† | 35.85 (7.23) | 0.12 | 28(34.1) | 22(26.8) | 32(39.0) | 0.39†† | 2.71 (1.93) | 0.17 |
| 1–2 times | 103(47.0) | 116(53.0) | | 34.50 (7.26) | | 66(31.4) | 63(30.0) | 81(38.6) | | 2.88 (2.05) | |
| 3–5 times | 22(40.7) | 32(59.3) | | 35.70 (8.30) | | 23(46.9) | 14(28.6) | 12(24.5) | | 2.16 (1.95) | |
| ≥6 | 16(41.0) | 23(59.0) | | 36.89 (7.76) | | 13(37.1) | 7(20.0) | 15(42.9) | | 2.69 (2.12) | |
| **Smoking habits** | | | | | | | | | | | |
| Smokers | 39(42.4) | 53(57.6) | 0.82† | 35.24 (6.39) | 0.99 | 34(40.0) | 21(24.7) | 30(35.3) | 0.58 | 2.64 (2.03) | 0.66 |
| Past smokers | 38(46.9) | 43(53.1) | | 35.22 (8.26) | | 30(37.5) | 20(25.0) | 30(37.5) | | 2.61 (2.16) | |
| Never smoked | 98(43.4) | 128(56.6) | | 35.16 (7.61) | | 66(31.3) | 65(30.8) | 80(37.9) | | 2.82 (1.97) | |
| **Alcohol consumption during the last month** | | | | | | | | | | | |
| Yes | 76(43.7) | 98(56.3) | 0.95† | 35.22 (7.29) | 0.94 | 42(25.1) | 53(31.7) | 72(43.1) | <0.01 | 3.12 (1.96) | <0.01 |
| No | 99(44.0) | 126(56.0) | | 35.16 (7.62) | | 88(42.1) | 53(25.4) | 68(32.5) | | 2.42 (2.02) | |
| **Frequency of alcohol consumption** | | | | | | | | | | | |
| More than 1 time/week | 10(43.5) | 13(56.5) | 0.41† | 35.67 (6.23) | 0.45 | 6(30.0) | 4(20.0) | 10(50.0) | 0.72†† | 3.25 (2.10) | 0.98 |
| 1 time/week | 20(42.6) | 27(57.4) | | 35.21 (6.99) | | 10(21.3) | 16(34.0) | 21(44.7) | | 3.13 (1.96) | |
| 2–3 times/ month | 26(52.0) | 24(48.0) | | 33.92 (7.83) | | 12(24.5) | 18(36.7) | 19(38.8) | | 3.04 (1.78) | |
| 1 time/month | 20(35.7) | 36(64.3) | | 36.17 (7.32) | | 14(26.4) | 16(30.2) | 23(43.4) | | 3.15 (2.09) | |
| **Exercise habits** | | | | | | | | | | | |
| Almost every day | 13(32.5) | 27(67.5) | 0.03† | 37.13 (8.12) | 0.19 | 13(33.3) | 13(33.3) | 13(33.3) | 0.19†† | 2.56 (1.80) | 0.21 |
| Sometimes per week | 41(40.6) | 60(59.4) | | 35.49 (7.38) | | 28(28.6) | 28(28.6) | 42(42.9) | | 2.97 (1.94) | |
| Sometimes per month | 40(48.2) | 43(51.8) | | 34.66 (6.78) | | 28(35.4) | 18(22.8) | 33(41.8) | | 2.96 (2.19) | |
| Never | 81(46.3) | 94(53.7) | | 34.82 (7.66) | | 61(38.1) | 47(29.4) | 52(32.5) | | 2.51 (2.03) | |

† Pearson chi-square.

††p value for trend was used because of linear by linear association.

age ($p = 0.04$), higher educational attainment ($p<0.01$), not receiving financial aid ($p = 0.03$) and higher monthly family income ($p<0.01$)–see Table 4. Specifically, fathers appeared to have higher on average functional HL score (M = 3.08, SD = 2.25) than mothers (M = 2.66, SD = 1.97). Almost one in two fathers (n = 30, 49.2%) had adequate functional HL, whereas one in three mothers (n = 110, 34.9%) demonstrated adequate functional HL. Furthermore, functional HL appeared higher across consecutive age groups, with the lowest NVS score observed among the 18–29 age group (M = 2.37, SD = 1.85) and the highest score being in the 35–39 age group (M = 3.07, SD = 2.09). However, in the age group of above 40 years of age, functional HL was slightly decreased to 2.98 (SD = 2.14). There was also a stepwise increase in functional HL score from a low of 1.91 (SD = 1.83) among parents with at most secondary education to 2.28 (SD = 2.05) among parents with college and 2.65 (SD = 2.01) university education to the highest score of 3.34 (SD = 1.94) among parents with postgraduate education. As shown on Table 4, while only one in five parents (22.2%) with at most secondary education had an adequate NVS score, the equivalent figure was higher in every consecutive educational group with the highest percentage recorded among parents with postgraduate education where nearly half (48.8%) had an adequate NVS score ($p$-value for trend$<0.01$). A social gradient in functional HL was also observed in terms of family income, in contrast to perceived HL which was not associated with income. There was an incremental increase in functional HL scores from 2.25 (SD = 1.97) in the lower income group to 2.75 (SD = 2.03), 3.21 (SD = 1.98), and 3.53 (SD = 1.88) among the highest income group. Expressed as a categorical variable, this corresponds to 27.2%, 36.4%, 51.4% and 51.2% with adequate functional HL across increasing income groups ($p$-value for trend$<0.01$). Additionally, almost one in three parents receiving financial aid (n = 51, 30.0%) had adequate functional HL compared to one in two parents among those not receiving a financial aid (n = 89, 43.2%), with the difference between the two groups being statistically significant ($p = 0.03$).

With regard to health-related behaviors, only alcohol consumption seems to be statistically associated with functional HL ($p<0.01$)–see Table 5. Interestingly, higher average scores were observed among parents who reported consuming alcohol in the past month, which nevertheless may be confounded by financial ability. No other meaningful associations were observed between functional HL and health-related behaviors.

## Multivariable analyses

In stepwise multivariable regression models, self-rated health status and number of children in the family were the only two variables which were statistically significantly associated with parental perceived HL ($F(2,382) = 4.592$, $p = 0.01$). Parental perceived HL was negatively associated with self-rated health status (b coeff = -1.03 per self-rated health status category increase, 95% CI -1.95 to -0.11) and positively associated with number of children in the family (b coeff = 1.00 per number of children in the family category increase, 95% CI 0.04 to 1.96) (Table 6).

With regard to functional HL, parental education, age, social status, self-rated health status, and alcohol consumption were statistically significantly associated with HL scores ($F(4,366) = 11.774$), $p<0.01$), whereby an average increase in functional HL scores was observed across increasing parental education (b coeff = 0.57 per educational level category increase, 95% CI 0.35 to 0.79), age (b coeff = 0.33 per age category increase, 95% CI 0.13 to 0.52), self-rated social status (b coeff = -0.31 per self-rated health status category increase, 95% CI -0.56 to -0.07), and alcohol consumption (b coeff = 0.51 per alcohol consumption category increase, 95% CI 0.10 to 0.91) during the last month (Table 6).

**Table 6. Multivariable analysis of perceived and functional HL after adjusting for all sociodemographic factors and health behaviors as estimated in multivariable logistic regression models.**

| Variables | B | 95% CI for B | | p-value |
|---|---|---|---|---|
| | | LL | UL | |
| **HLS-EU-Q47** | | | | |
| Model† | | | | |
| Self-rated health status | -1.028 | -1.95 | -0.11 | 0.03 |
| No of children in the family | 1.005 | 0.04 | 1.96 | 0.04 |
| **NVS** | | | | |
| Model † | | | | |
| Education | 0.573 | 0.35 | 0.79 | <0.01 |
| Age | 0.326 | 0.13 | 0.52 | <0.01 |
| Social status | -0.312 | -0.56 | -0.07 | 0.01 |
| Alcohol consumption | 0.506 | 0.10 | 0.91 | 0.01 |

Note. Model = stepwise method in SPSS Statistics; B = unstandardized regression coefficient; CI = confidence interval; LL = lower limit; UL = upper limit.

†Model adjusted for child's gender, child's age, parental gender, parental age, place of residence, nationality, marital status, educational attainment, financial aid, financial difficulties, employment status, family income, social status, number of children in the family, exercise habits, smoking habits, alcohol consumption, doctor visits.

## Discussion

The results of the current study have shown that a high proportion of parents in Cyprus may have inadequate or problematic HL, according to both self-assessed and performance-based HL tools. Higher educational attainment, higher number of children, better health self-rated status, and more frequent exercise were statistically significantly associated with higher scores of parental perceived HL, whereas male gender, older age, higher educational attainment, not receiving financial aid, higher monthly family income, and alcohol consumption were statistically significantly associated with higher scores of parental functional HL. The combination of self-rated health status and number of children in the family were observed as important predictors of parental perceived HL, whereas the combination of self-rated health status, parental education, age, social status, and alcohol consumption as important predictors of parental functional HL.

In the current study, mothers appeared to have higher perceived HL score with almost one in four of them demonstrate excellent perceived HL, but with not any statistically significant association. Similarly, a recent study [26] have not found any statistically significant association between gender and perceived HL in a sample of Cypriot and Greek population. However, male gender was significantly associated with higher scores of functional HL. This finding could be explained by the results of international large-scales tests which indicate that males have better mathematic performance and achievement [31], a main skill assessed in NVS tool.

The results of the current study indicate that being older is a significant predictor for having higher performance-based HL but not self-assessed HL. On the contrary, in a previous study [32], older age was associated with lower performance-based HL and not with self-assessed HL, which is consisted with the findings of a systematic review [43] that reported older age to be strongly associated with limited functional HL. In the same line, in a sample of Cypriot and Greek population, higher proportions of people with limited perceived HL were found among those who were older [26]. However, in the current study the participants were parents with children under the age of eighteen years old. Hence, the sample of this study was younger, with only 2.1% of them being above 49 years old.

Parents with other nationality than Greek Cypriot, demonstrated lower self-assessed HL levels than Cypriot parents, with more than one in two parents with other nationality appear problematic HL. In contrast with the perceived HL findings, NVS score was higher in parents without a Greek Cypriot nationality. Despite the differences between the two groups of the current study were not statistically significant, several previous studies support that people with other nationality than their country of residence had limited HL [33–35]. This may be since, in the current study, nearly half of the parents with other nationality than Greek Cypriot had a Greek nationality (n = 15, 41.7%) and hence, speak the same language as Greek Cypriot parents. Language barrier, in the other studies, seemed to be one of the main factors in relation to people with other nationalities which led to limited HL, which was not a common characteristic of the parental sample included in the current study.

Educational attainment was found to be a significant predictor of both perceived and functional HL. In accordance with these results, a study conducted in Denmark [32] found a significant association between education and HL in people aged 16–65 years, based on the HLS-EU-Q47 and the performance-based tool Health Activities and Literacy Scale (HALS). A study conducted in a European sample [37], and a study conducted in Greek and Cypriot participants [26], also agree that higher education is significantly associated with higher scores in perceived HL. However, in the current study the pattern observed with functional HL and education was linear unlike perceived HL, whereby the postgraduate group did not demonstrate the highest score. This can lead to the assumption that perceived HL may be influenced by individual's expectations. Thus, parents with the highest education did not rate their perceived HL as high, possibly due to their higher expectations, as supported by the observation that they performed better on average in terms of the functional HL measure. Additionally, individual's expectations seemed to influence the parental perception of social status. In multivariable analyses, parents who rated higher their social status showed to have lower functional HL score after controlling for other sociodemographic and health behavior variables, including educational level, which had a positive relationship with functional HL.

Similarly, parental functional HL demonstrated a stepwise increase in terms of family income, whereas perceived HL was not associated with family income. This also supports that perceived HL may be differentially influenced by expectations. While participants in the highest income group did not rate their perceived HL as high, they tend to perform better in the functional HL measure. Equivalently, people in lower income categories tend to rate their HL higher but have on average lower functional HL scores. In accordance, reported facing financial difficulties was observed in parents with higher self-assessed HL, even without a statistically significant association, whereas receiving financial aid was a significant predictor of parental higher performance-based HL. Similarly, in the first European comparative survey using the HLS-EU-Q conducted in 2011 between eight European countries [37], financial problems showed to be a significant predictor of lower levels of self-assessed HL among eight European countries. On the other hand, even without a statistically significant difference, increased social status was associated with perceived HL, indicating that objective indicators such as, educational attainment and income seem to be capturing the social gradient in perceived HL better than subjective indicators such as, social status.

Interestingly, alcohol consumption found to be a significant predictor of higher functional HL, in the current study. During the last years, a new HL concept, named 'distributed health literacy' has emerged focusing on the intersection of HL in the social context, supporting that an individual's HL can be influenced by his/her social environment and interactions with others. Social engagement was a common variable measured to represent the social context variable [44,45]. Alcohol consumption is often related to social engagement and interactions. Therefore, it could be assumed that according with this study's results, in which participants

with alcohol consumption more than three times per week was only 4.5%, alcohol consumption for social reasons (possibly more social interaction and discussions with others increased their HL) seem to be related positively with functional HL.

Furthermore, the use of health care services seemed to not be associated with perceived or functional HL. However, it was observed that those who had visited a doctor during the last year for six or more times demonstrated higher perceived HL levels than those who visited a doctor less times. Nevertheless, the association was not statistically significant. These findings were not congruent with the findings of a large European study [37], in which the highest proportion of limited HL was observed for participants reported six or more doctor visits in the last year. Parents with increased doctor visits, in the current study, may perceive that their HL skills had increased in the three domains of health, as a result of the frequent use of healthcare services and communication with their doctor and other health professionals.

Most of the studies exploring HL in literature examine only one aspect of HL. The current study assessed parental HL, using both perceived (self-assessed) and functional (performance-based) tools. The results demonstrated no correlation between the two measurement tools of HL, with increased self-assessed HL score being observed among participants with high likelihood of limited performance-based HL, and decreased performance-based HL score being observed among participants in the excellent group of self-assessed HL. It should be pointed out that perceived and functional HL are two different constructs of HL. Perceived HL describes the way in which people understand their own skills and abilities in accessing, understanding, appraising, and applying health information. Perceived HL may be influenced by the individual's self-confidence, self-perception, and culture. It can be assumed that social desirability may influence perceived HL, as parents with limited performance-based HL rated higher their perceived HL. On the other hand, functional HL refers to a more objective measurement of HL in regard to people's practical skills, such as reading, understanding, and numeracy. Moreover, functional HL measures are task- or performance-based. Thus, subjective assessment of HL (HLS-EU-Q47) may not necessarily be indicative of the actual performance in a specific HL-related task due to the restricted range of HL skills covered by the NVS (i.e. nutritional label). Hence, self-assessment of HL does not necessarily correspond to the performance in tasks measuring specific HL skills. Additionally, this finding supports not only the fact that HLS-EU-Q47 is a subjective measure of HL, while NVS is an objective, but also that HLS-EU-Q47 measures a broader spectrum of HL, including four components (accessing, understanding, appraising, and applying) in the three health dimensions (healthcare, disease prevention, health promotion), while NVS measures specific skills (reading, numeracy) of HL. Interestingly though, both measures captured the expected gradient with educational attainment and other socio-demographic characteristics. Concurrent use of perceived and functional HL measures may provide a better assessment of HL. Using both subjective and objective tools of HL, allows the exploration of individual's HL in a broader and more holistic manner. Future studies should be conducted to assess HL using both subjective and objective tools to gain a broader understanding of an individual's HL and to further examine the correlation between them.

The current study has several limitations. First, a stratified sampling may be more appropriate rather than the convenience sampling used in this study which may lead to selection bias. However, convenience sampling was more feasible during the Covid-19 pandemic period, in which data collection took place, because of the restrictions in paediatric clinical settings. Additionally, while selection bias cannot be excluded due to both convenience sampling approach and voluntary nature of participation, consecutive sampling of potential participants according to a visitation schedule was followed, as per the study protocol, in order to minimize any sampling bias originating from the researcher. Additionally, a high percentage of

participants were University graduates with a bachelor or postgraduate degree (n = 286, 68.7%). While selection bias towards participants with higher educational attainment may be possible, the demographic composition of the sample does not appear so atypical to a national sample. According to the official Perinatal Health Report of the Ministry of Health [46], the percentage of women with post-secondary education giving birth in Cyprus was 74% in 2015. However, the official report does not further break down this figure to college and university level education, as in the current study. Since the participants included women who will had 5-6-year-old children at the time of the current study, it could support the representativeness of the sample. Moreover, the cross-sectional nature of the design, disallows any causal conclusions. Therefore, a longitudinal design should be implemented in future studies. Moreover, the analytical approach taken, regarding stepwise multivariable analysis, does not allow any inference in terms of predictors and outcomes of HL, as a bigger sample size would be required to delineate these structural relationships, which was beyond the scope of this work. A much bigger sample would also be required for modelling HL against a big set of variables with mutual adjustments. Thus, a more pragmatic, rather than theoretically-oriented, analytical strategy was preferred, whereby firstly, the observed differences in HL were assessed according to each variable (main aim) and then, a stepwise multivariable approach was followed with perceived/functional HL as dependent variables and all other variables as independent variables, also avoiding collinearity between them, with the aim to identify the minimum set of variables associated with HL (secondary aim). Finally, data collection took place during the Covid 19 pandemic. This was a period where people were more exposed to digital technology. This may influence their skills in searching and accessing health information. People were also more exposed to health information through media, which may influence their understanding and in turn, applying of health information. Surprisingly though, previous study [36] conducted in Cypriot population showed similar results in HL levels among 300 health care users at the General Hospital of Limassol in Cyprus. Specifically, half of the participants perceived themselves to have inadequate or problematic (50.7%) HL, a similar proportion observed in the current study (42.6%). While it is risky to draw comparison across different samples, this may suggest that conditions during the two-year period of data collection in the pandemic did not significantly change HL levels of the population. On the other hand, the pandemic period may have had an impact on health behaviors. Restrictions imposed during the pandemic period may limit the exercise habits of people at the gym, but it may increase the opportunity to exercise at home or outdoors as a way to escape from isolation. In fact, there is some evidence from Cyprus to suggest that overall physical activity levels did not change much in this period, even though the data is self-reported. Based on the same study, smoking may have increased in this period, but alcohol consumption may have decreased due to restrictions which limit social interactions and visits to restaurants or bars [47]. Sociodemographic characteristics are probably the same.

Further research is needed in this specific population using both self-reported and performance-based tools. The lack of correlation between perceived and functional HL highlights the importance in assessing both in parallel to gain a holistic insight of HL level in populations. Personal expectations, social and cultural context may differentially influence perceived HL. Future studies should consider the impact of these aspects in HL. This can contribute to the development of a wider range of knowledge of the factors influencing HL. Additionally, the current study has identified several risk factors which can be used for the early recognition of parents with limited HL by healthcare practitioners and inform the development of interventions addressing parental limited HL. Educational interventions should prioritize parents with identified risk factors in terms of gender, age, educational attainment, family income, and number of children in the family. Limited parental HL observed in the current study also

highlights the need to consider and include HL competencies for all three domains of health (healthcare, disease prevention, health promotion) when designing and developing pediatric health services, such as providing HL-appropriate material in finding and assessing reliable health information or navigating in the pediatric center services. Health-literate pediatric healthcare organizations, centers, or hospitals, adapted to parental HL needs, could assist parents by minimizing their difficulties in engaging and navigating with those centers, communicating with healthcare practitioners more effectively, taking informative health-decisions and acting appropriately regarding their child's health.

## Conclusion

In order to maintain good health and wellbeing in children, parents need to obtain and understand health information from a variety of sources, such as healthcare practitioners and online health sources, to appraise the quality of this information, and to take decisions and act effectively for their child's health and wellbeing. The skills needed to deal with all these aspects can be referred to as HL. Most studies examine separately perceived HL and functional HL. This study, not only identifies parental characteristics associated with limited HL, but also highlights the importance of assessing both types of HL (self-assessed and performance-based) to establish a comprehensive assessment of parental HL.

A substantial percentage of Cypriot parents, in the current study, have low HL, and this conclusion is consistent irrespective of whether a perceived or functional measure is used. Nevertheless, from a large number of variables investigated, indicators of social gradient were found to be statistically significantly associated with HL. While self-rated health was associated with HL, the association between HL and health-related behaviours did not appear to show a clear pattern. Findings suggest that some parental characteristics may play an important role as predictors of their limited HL. This could imply important information in identifying parents at risk for limited HL levels to appropriately intervene and improve not only parental HL, but also consequently health and wellbeing in pediatric population.

## Supporting information

**S1 Table. STROBE checklist.**
(DOCX)

**S1 Appendix. NVS tool.**
(PDF)

**S2 Appendix. HLS-EU-Q47 tool.**
(PDF)

**S3 Appendix. Sociodemographic tool.**
(PDF)

## Acknowledgments

We would like to thank the three primary pediatric centers which provided access for the collection of the data during the covid-19 pandemic.

## Author Contributions

**Conceptualization:** Christiana Nicolaou.

**Data curation:** Joanna Menikou, Nicos Middleton.

**Formal analysis:** Joanna Menikou.

**Methodology:** Joanna Menikou, Nicos Middleton, Christiana Nicolaou.

**Supervision:** Nicos Middleton, Evridiki Papastavrou, Christiana Nicolaou.

**Writing – original draft:** Joanna Menikou.

**Writing – review & editing:** Joanna Menikou, Nicos Middleton, Evridiki Papastavrou, Christiana Nicolaou.

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
