## [Decision Letter · Decision Letter 0]

12 Jul 2023

PONE-D-23-10965Assessing perceived and functional health literacy among parents in Cyprus: a cross-sectional studyPLOS ONE

Dear Dr. Menikou,

Thank you for submitting your manuscript to PLOS ONE. After careful consideration, we feel that it has merit but does not fully meet PLOS ONE’s publication criteria as it currently stands. Therefore, we invite you to submit a revised version of the manuscript that addresses the points raised during the review process.

We look forward to receiving your revised manuscript.

Kind regards,

Iyn-Hyang Lee, Ph.D.

Academic Editor

PLOS ONE

Journal Requirements:

2. We noted in your submission details that a portion of your manuscript may have been presented or published elsewhere. Please clarify whether this [conference proceeding or publication] was peer-reviewed and formally published. If this work was previously peer-reviewed and published, in the cover letter please provide the reason that this work does not constitute dual publication and should be included in the current manuscript.

4. We note you have included a table to which you do not refer in the text of your manuscript. Please ensure that you refer to Table 6 in your text; if accepted, production will need this reference to link the reader to the Table.

Additional Editor Comments:

Please clarify why your study has been done, and what questions are being answered. This is a very important issue that could determine the value of this study. I hope that the authors can sufficiently improve this point in their revised manuscript.

Reviewers' comments:

Reviewer's Responses to Questions

**Comments to the Author**

1. Is the manuscript technically sound, and do the data support the conclusions?

Reviewer #1: Partly

Reviewer #2: Yes

2. Has the statistical analysis been performed appropriately and rigorously? 

Reviewer #1: I Don't Know

Reviewer #2: Yes

3. Have the authors made all data underlying the findings in their manuscript fully available?

Reviewer #1: No

Reviewer #2: Yes

4. Is the manuscript presented in an intelligible fashion and written in standard English?

Reviewer #1: Yes

Reviewer #2: Yes

5. Review Comments to the Author

Reviewer #1: The study has some strengths, including the collection of HL data from two sources. However, it's not made completely clear why this study is needed. What service or practitioner changes could result?

The study used a convenience sample, based on parents accompanying child to the practice. There were research visits 2-3 times per week, at different times, which is good. Recruitment was undertaken by 1 researcher only. Did they work to a study protocol? Were consecutive attendees approached? If not, sampling biases are possible.

It's good that the authors included a sample size calculation based on an aniticipated SD of the HL scores. But it’s unclear why the paper also talks about the size of sample needed to detect a certain effect - that doesn't seem relevant to a survey design.

It used well-established HL tools, which is a strength.

The authors state: “Stepwise multiple linear regression analyses were used to identify the strongest associations between parental perceived HL or functional HL scores with all predictor variables: parental sociodemographic characteristics and parental health behaviours after mutually adjusting for each other.” However, there are aspects of this analysis plan that are unclear.

First, is the sample big enough to accommodate the analysis of predictors? It may need a bigger sample than for estimating the mean and variation.

Second, what model is being assumed here, by incorporating both the socio-demographics and the health behaviours? How does HL fit in with this? What are the predictor variables and what are the outcome variables? As it stands, the analysis includes both sets of variables. There have been mutual adjustments of the sets of variables, which may be appropriate but needs explaining in terms of the model being evaluated.

The authors report problems with participants answering Qs 3 and 4 of the HL scale. But given the reported text in the manuscript, I cannot answer them – not enough information is given in the questions!

Item 4 (‘If you usually eat 2.500 calories in a day, what percentage of your daily value of calories will you be eating if you eat one serving?”) (n=255, 61.3%). What is a "serving"?

Item 3 (‘Your doctor advises you to reduce the amount of saturated fat in your diet. You usually have 42g of saturated fat each day, which includes one serving of ice cream. If you stop eating ice cream, how many grams of saturated fat would you be consuming each day?”). Where is it explained the number of g of saturared fat in one ice cream serving?

“Finally, a high percentage of participants were University graduates with a bachelor or postgraduate degree (n=286, 68.7%).” This probably makes the sample nationally untypical, but by how much?

Reviewer #2: Review Report for PLOS ONE PONE-D-23-10965

Assessing Perceived and functional health literacy among Parents in Cyprus: a cross-sectional Study

Joanna Menikou, PhDc., Nicos Middleton, Evridiki Papastavrou, Christiana Nicolaou

The authors wrote an interesting article, with a good research idea, and already well-written.

Below are my suggestions.

1. Abstract

• It will be more informative if the result part that explains the multivariate result could also show the direction of the association (for example; with better educational attainment, the higher number of children will have better perceived HL)

2. Background

• The association between sociodemographics and Health Literacy (functional and perceived) should be given profs from previous studies.

3. Method

• Need to provide the justification that the condition in two years period of data collection didn’t change significantly for the Independent and dependent variables

• Should provide the result of validity and reliability analysis for Perceived and Functional HL, and give interpretation about the result if compared with previous studies.

4. Result

• Should provide an explanation of why there is no correlation between the Perceived and Functional HL. Is it the same phenomenon as the previous study conducted before or not?

• The P-values shown in tables should be in the same format, use < 0.05/<0.01 rather than 0.000

5. Discussion and Conclusion are clear and well written

6. PLOS authors have the option to publish the peer review history of their article (what does this mean?). If published, this will include your full peer review and any attached files.

Reviewer #1: **Yes: **Peter Knapp, University of York

Reviewer #2: **Yes: **Junaidi Budi Prihanto

---

## [Author Response · Author response to Decision Letter 0]

29 Aug 2023

Editor Comments:

1. Please clarify why your study has been done, and what questions are being answered.

Response: 

We thank the editor for giving us the opportunity to clarify the significance of the study. Changes have been made in the text as follows:

Introduction: 

- Added in p.g. 3 (lines 57-68):

‘…Parents often need the competencies to find and assess reliable health information for a variety of children’s health issues, to effectively communicate with healthcare professionals, manage health conditions of their child, respond to symptoms, and deal with acute illness….. In addition, two earlier systematic reviews [11, 12] and a more recent scoping review [13] revealed that limited parental HL was associated with negative health behaviors and health outcomes in children. Therefore, a primary issue in promoting health, preventing illness, and providing effective healthcare in children could be the enhancement of parental HL.’

- Added in p.g. 5 (lines 104-108):

‘Parents are the ones who are responsible for their children’s health. Identifying parents with limited HL is vital in ensuring best practices regarding health promotion, disease prevention, and healthcare in childhood population. Furthermore, assessing both types of HL (self-assessed and performance-based) may lead in a more holistic and comprehensive assessment of parental HL.’

Conclusion: 

Added in p.g. 32 (lines 623-630: ‘In order to maintain good health and wellbeing in children, parents need to obtain and understand health information from a variety of sources, such as healthcare practitioners and online health sources, to appraise the quality of this information, and to take decisions and act effectively for their child’s health and wellbeing. The skills needed to deal with all these aspects can be referred to as HL. Most studies examine separately perceived HL and functional HL. This study, not only identifies parental characteristics associated with limited HL, but also highlights the importance of assessing both types of HL (self-assessed and performance-based) to establish a comprehensive assessment of parental HL.’

In regard to the second part of the comment about the questions that have been answered in the study, previously in the text (‘Introduction’ section) was mentioned that: ‘The primary aim of the current study was to examine both self-assessed HL and performance-based HL among Cypriot parents. Additionally, the study explored the extent to which sociodemographic characteristics (such as educational attainment, financial status) and health-related behaviours (such as smoking, exercise habits) are potential predictors of parental perceived HL and functional HL, in order to identify a set of characteristics of a parent with limited HL.’ We have changed it as follows (lines 115-121):

‘… Therefore, an additional objective was to identify a set of characteristics of a parent with limited HL. Hence, the following research questions were examined: 

1. Are sociodemographic characteristics (such as educational attainment, financial status) potential predictors of parental perceived HL and functional HL?

2. Are health-related behaviours (such as smoking, exercise habits) potential predictors of parental perceived HL and functional HL?’

Reviewers’ comments:

Reviewer #1

1. “It’s not made completely clear why this study is needed. What service or practitioner changes could result?”

Response: Regarding the first part of the comment: ‘It’s not made completely clear why this study is needed’, the reply is the same as in the ‘Additional Editor’s Comment’ above.

Regarding the second part of the comment: ‘What service or practitioner changes could result?’, the following text is added in the ‘Discussion’ (p.g. 31-32, lines 606-619):

‘Additionally, the current study has identified risk factors which can be used for the early recognition of parents with limited HL by healthcare practitioners and inform the development of interventions addressing parental limited HL. Educational interventions should prioritize parents with identified risk factors in terms of gender, age, educational attainment, family income, and number of children in the family. Limited parental HL observed in the current study also highlights the need to consider and include HL competencies for all three domains of health (healthcare, disease prevention, health promotion) when designing and developing pediatric health services, such as providing HL-appropriate material in finding and assessing reliable health information, or navigating in the pediatric center services. Health-literate pediatric healthcare organizations, centers, or hospitals, adapted to parental HL needs could assist parents by minimizing their difficulties in engaging and navigating with those centers, communicating with healthcare practitioners more effectively, taking informative health-decisions and acting appropriately regarding their child’s health.’

2. “The study used a convenience sample, based on parents accompanying child to practice. There were research visits 2-3 times per week, at different times, which is good. Recruitment was undertaken by 1 researcher only. Did they work to a study protocol? Were consecutive attendees approached? If not, sampling biases are possible.”

Response: We would like to thank the reviewer for this comment which gives us the opportunity to clarify. Indeed, while recruitment was undertaken by one researcher, the process was regulated by a study protocol, which among others called for a consecutive sample of visitors who fulfilled the inclusion criteria be approached. This was clarified in the ‘Materials and methods’ section (‘Data collection procedure’) of the manuscript (p.g. 7, lines 155-161), as follows:

“Recruitment of potential participants was performed according to a study protocol, detailing the eligibility criteria for inclusion in the study, data collection process (i.e. data collection at different days and times each week) and sampling. In terms of sampling, a consecutive sample of practice visitors who fulfilled the eligibility criteria were approached and invited to participate in the study. A pilot study was also undertaken prior to the main study to explore both the feasibility of the study protocol as well as the measurement reliability of the questionnaire in the specific population.”

Nevertheless, while the study protocol minimizes any selection bias originating from the researcher, selection bias cannot be excluded both due to the convenience sampling approach and voluntary nature of participation. This is added as a limitation of the study in the ‘Discussion’ section (p.g. 29, lines 560-563) as follows:

“Additionally, while selection bias cannot be excluded due to convenience sampling approach and voluntary nature of participation, consecutive sampling of potential participants according to a visitation schedule was followed, as per the study protocol, in order to minimize any sampling bias originating from the researcher.”

3. “It’s good that the authors included a sample size calculation based on anticipated SD of the HL scores. But it’s unclear why the paper also talks about the size of sample needed to detect a certain effect – that doesn’t seem relevant to a survey design.”

Response: To determine the minimum required sample size, we performed precision analysis for the main variables of interest as appropriate for a survey design. However, since one of the research questions of the study was to assess differences in HL according to parental sociodemographic characteristics and health behaviors, power analysis under different scenarios/variables was also undertaken, as a supplementary approach, in order to determine the minimum difference needed to be observed to be statistically significant. Following the reviewer’s suggestion, we now removed any reference to power analysis to avoid any confusion with the readers.

4. “The authors’ state: ‘Stepwise multiple linear regression analyses were used to identify the strongest associations between parental perceived HL or functional HL scores with all predictor variables: parental sociodemographic characteristics and parental health behaviors after mutually adjusting for each other.’ However, there are aspects of this analysis plan that are unclear.

First, is the sample big enough to accommodate the analysis of predictors? It may need a bigger sample than for estimating the mean and variation.

Second, what model is being assumed here, by incorporating both the socio-demographics and the health behaviors? How does HL fit in with this? What are the predictor variables and what are the outcome variables? As it stands, the analysis includes both sets of variables. There have been mutual adjustments of the sets of variables, which may be appropriate but needs explaining in terms of the model being evaluated.”

Response: The reviewer is right in pointing out that a much bigger sample size would be required to delineate these structural relationships (e.g. structural equation modelling) which was beyond the scope of this work. A much bigger sample would also be required for modelling HL against a big set of variables with mutual adjustments. Thus, a more pragmatic, rather than theoretically-oriented, analytical strategy was preferred whereby we (a) first assessed the observed differences in HL according to each variable (main aim) and (b) followed stepwise multivariable approach with perceived/functional HL as dependent variables and all other variables as independent variables, also avoiding collinearity between them, with the aim to identify the minimum set of variables associated with HL (secondary aim). This was further clarified in the ‘Methods’ (p.g.11, lines 251-252) while we also explained in the ‘Limitations’ (p.g.30, lines 579-589) that the analytical approach taken does not allow any inference in terms of predictors and outcomes of HL.

5. “The authors’ report problems with participants answering Qs 3 and 4 of the HL scale. But given the reported text in the manuscript, I cannot answer them – not enough information is given in the questions!”

Response: The NVS questionnaire is added as Supporting Information.

6. “Finally, a high percentage of participants were University graduates with a bachelor or postgraduate degree (n=286, 68.7%). This probably makes the sample nationally atypical, but by how much?”

Response: Thanks for pointing this out as it actually gives us the opportunity to clarify. Previously, in the ‘Discussion’ section we had mentioned the following: ‘Additionally, a high percentage of participants were University graduates with a bachelor or postgraduate degree (n=286, 68.7%), which is not in agreement with the Cyprus population census 2011, in which the proportion of people in the participants’ age groups who had university degree (bachelor or postgraduate) was approximately 27.8%. This concludes a possible selection bias towards higher educational attainment.’ 

There was a mistake in this reference, since the percentage 27.8% with university education refers to the total population and not the age-group addressed in this study. Unfortunately, information from the latest 2021 census of the Cypriot population is not available yet. However, according to the official Perinatal Health Report of the Ministry of Health, the percentage of women with post-secondary education giving birth in Cyprus was 74% (in 2015). This is a better statistic to compare the representativeness of our sample, since these are women who will have had 5–6-year-old children at the time of the survey. Unfortunately, the official report does not further break down this figure to college and university level education, like in our study. The percentage of university (bachelor or postgraduate) education recorded here might still be elevated compared to national statistics. Therefore, while selection bias towards participants with higher educational attainment cannot be ruled out, the demographic composition of the sample does not appear so atypical to a national sample. We have now corrected the previous mention and provided this analysis instead in the ‘Discussion’ section (p.g.30, lines 565-572).

Reviewer #2

1. Abstract

• It will be more informative if the result part that explains the multivariate result could also show the direction of the association.

Response: The direction of associations for perceived and functional HL have been added in the ‘Results’ section of the ‘Abstract’ (p.g.2, lines 35-40).

2. Background

• The association between sociodemographics and HL (perceived and functional) should be given profs from previous studies.

Response: It was mentioned in the ‘Materials and methods’ section (p.g. 10, lines 213-215) that ‘Participants were also asked to provide basic sociodemographic information, including age, gender, marital status, nationality, place of residence (urban vs rural), and other, expected to demonstrate an association with HL based on previous studies [28-29].’

After the reviewer’s comment, the following text is also added in the ‘Introduction’ (p.g. 5, lines 113-115):

‘Several sociodemographic characteristics have been associated with HL in adult population, such as gender [28], age [23, 29], nationality [30, 31, 32], and educational attainment [23, 29, 34].’

3. Method

• Need to provide the justification that the condition in two years period of data collection didn’t change significantly for the independent and dependent variables.

Response: Indeed, the data collection was performed over a two-year period, which largely coincided with the pandemic. Following the reviewer’s suggestion, we have discussed this matter, in ‘Discussion’ section (p.g. 30, lines 587-604) as follows: 

‘This was a period where people were more exposed to digital technology. This may influence their skills in searching and accessing health information. People were also more exposed to health information through media, which may influence their understanding and in turn, applying of health information. Surprisingly though, previous study [28] conducted in Cypriot population showed similar results in HL levels among 300 health care users at the General Hospital of Limassol in Cyprus. Specifically, half of the participants perceived themselves to have inadequate or problematic (50.7%) HL, a similar proportion observed in the current study (42.6%). While it is risky to draw comparison across different samples, this may suggest that conditions during the two-year period of data collection in the pandemic did not significantly change HL levels of the population. On the other hand, the pandemic period may have had an impact on health behaviors. Restrictions imposed during the pandemic period may limit the exercise habits of people at the gym, but it may increase the opportunity to exercise at home or outdoors as a way to escape from isolation. In fact, there is some evidence from Cyprus to suggest that overall physical activity levels did not change much in this period, even though the data is self-reported. Based on the same study, smoking may have increased in this period but alcohol consumption may have decreased due to restrictions which limit social interactions and visits to restaurants or bars [44]. Sociodemographic characteristics are probably the same.’

• Should provide the result of validity and reliability analysis for Perceived and Functional HL, and give interpretation about the result if compared with previous studies.

Response: Regarding perceived HL (HLS-EU-Q47): We had previously mentioned in the manuscript that ‘a pilot validity assessment of the Greek translated version of the tool showed good metric properties’ amongst a Cypriot general population sample.’ We have now provided more details since previous reference was generic, as follows (in ‘Assessment of outcome variables’ – ‘Perceived HL’, p.g. 9, lines 193-201): 

‘Specifically, the study supported the construct factor validity of the tool. Furthermore, the study showed that the tool was able to capture the expected social gradient in HL by indicators of socio-economic disadvantage, including educational attainment, further supporting its criterion-based validity. In terms of the dimensionality, the recommendation was to 4 skills sub-scales or 3 health domain sub-scales, rather than across all 12 sub-scales of the 4X3 HL conceptual model. As this is not an uncommon finding in the literature, this conceptualization was preferred here to calculate aggregate scores, only for descriptive purposes. Only the overall score, with a high level of internal consistency, as determined by a Cronbach’s a of 0.958, was used in the analysis’.

Regarding functional HL (NVS): In terms of the NVS, the observed relationship with HLS-EU-Q47 was intended to function as a form of concurrent criterion-related validity for the tool. As discussed though in the manuscript (‘Results’ – ‘Perceived and functional HL’-, p.g. 16-17, lines 335-339) the low observed correlation between perceived HL and functional HL raises questions since, as we have now added: ‘The low observed correlation between perceived HL and functional HL might suggest that concurrent validity is limited between the two tools (NVS and HLS-EU-Q47). However, subjective assessment of HL (HLS-EU-Q47) may not necessarily be indicative of the actual performance in a specific HL-related task due to the restricted range of HL skills covered by the NVS (i.e nutritional label).’

4. Result

• Should provide an explanation of why there is no correlation between the Perceived and Functional HL. Is it the same phenomenon as the previous study conducted before or not?

Response: This is a very interesting point and indeed finding of the study. As also explained in response to the previous point, the correlation between the two measures function as a form of concurrent validity. The lack of correlation raises questions. We had previously offered some explanations for this phenomenon (in ‘Discussion’ section, p.g. 28, lines 527-532), but it is reach inferences since studies do not tend to concurrently use perceived and functional measure of HL. One possibility that is at least partly supported by the discordance observed at opposite ends of the HL continuum is that social desirability may be at play, for example, higher perceived HL among participants with otherwise low functional HL. However, other explanations are also likely, so we have now expanded on the ‘Discussion’ (p.g. 28-29, lines 532-552) as follows:

‘It should be pointed out that perceived and functional HL are two different constructs of HL. Perceived HL describes the way in which people understand their own skills and abilities in accessing, understanding, appraising, and applying health information. Perceived HL may be influenced by the individual’s self-confidence, self-perception, and culture….On the other hand, functional HL refers to a more objective measurement of HL in regard to people’s practical skills, such as reading, understanding, and numeracy. Moreover, functional HL measures are task- or performance-based. Thus, subjective assessment of HL (HLS-EU-Q47) may not necessarily be indicative of the actual performance in a specific HL-related task due to the restricted range of HL skills covered by the NVS (i.e. nutritional label). …….Interestingly though, both measures captured the expected gradient with educational attainment and other socio-demographic characteristics. Concurrent use of perceived and functional HL measures may provide a better assessment of HL.’

• The p-values shown in tables should be in the same format, use <0.05/<0.01 rather than 0.000.

Response: We thank the reviewer for this observation. P-values have been changed to <0.01 both in tables and text.

---

## [Decision Letter · Decision Letter 1]

25 Sep 2023

Assessing perceived and functional health literacy among parents in Cyprus: a cross-sectional study

PONE-D-23-10965R1

Dear Dr. Menikou,

We’re pleased to inform you that your manuscript has been judged scientifically suitable for publication and will be formally accepted for publication once it meets all outstanding technical requirements.

Kind regards,

Iyn-Hyang Lee, Ph.D.

Academic Editor

PLOS ONE

Additional Editor Comments (optional):

Reviewers' comments:

Reviewer's Responses to Questions

**Comments to the Author**

1. If the authors have adequately addressed your comments raised in a previous round of review and you feel that this manuscript is now acceptable for publication, you may indicate that here to bypass the “Comments to the Author” section, enter your conflict of interest statement in the “Confidential to Editor” section, and submit your "Accept" recommendation.

Reviewer #1: All comments have been addressed

Reviewer #2: All comments have been addressed

2. Is the manuscript technically sound, and do the data support the conclusions?

Reviewer #1: Yes

Reviewer #2: Yes

3. Has the statistical analysis been performed appropriately and rigorously? 

Reviewer #1: Yes

Reviewer #2: Yes

4. Have the authors made all data underlying the findings in their manuscript fully available?

Reviewer #1: Yes

Reviewer #2: Yes

5. Is the manuscript presented in an intelligible fashion and written in standard English?

Reviewer #1: Yes

Reviewer #2: Yes

6. Review Comments to the Author

Reviewer #1: (No Response)

Reviewer #2: I want to thank the author for all of their efforts in revising the essay in response to the initial round of reviewers' suggestions

7. PLOS authors have the option to publish the peer review history of their article (what does this mean?). If published, this will include your full peer review and any attached files.

Reviewer #1: **Yes: **Dr Peter Knapp, University of York, UK

Reviewer #2: **Yes: **Junaidi Budi Prihanto

---

## [Editor Report · Acceptance letter]

2 Oct 2023

PONE-D-23-10965R1 

Assessing perceived and functional health literacy among parents in Cyprus: a cross-sectional study 

Dear Dr. Menikou:

I'm pleased to inform you that your manuscript has been deemed suitable for publication in PLOS ONE. Congratulations! Your manuscript is now with our production department. 

Kind regards, 

on behalf of

Prof. Iyn-Hyang Lee 

Academic Editor

PLOS ONE